# On the Relationship between Pro-Environmental Behavior, Experienced Monetary Costs, and Psychological Gains

**Mathias Zannakis [1],\***[ID], **Sverker Molander [2]**[ID] **and Lars-Olof Johansson [3]**

[1] Department of Social and Behavioural Studies, University West, 461 86 Trollhättan, Sweden
[2] Department of Technology Management and Economics, Division of Environmental Systems Analysis, Chalmers University of Technology, 412 96 Gothenburg, Sweden; sverker.molander@chalmers.se
[3] Department of Psychology, University of Gothenburg, 413 14 Gothenburg, Sweden; lars-olof.johansson@psy.gu.se
\* Correspondence: mathias.zannakis@hv.se; Tel.: +46-0522-223-829

**Abstract:** Drawing on the emerging scarcity, abundance, and sufficiency (SAS) framework, this study explores how various consumer behaviors with potential environmental impacts relate to subjective evaluations of psychological resources such as economic resources, time, social networks, and emotional support. Assuming that individuals may "trade" the costs and efforts of green consumption, including the buying of eco-labeled goods, altered eating habits, and choice of transportation mode, against such psychological resources, we investigate the relationships between green consumer choices and resource evaluations using hierarchical regression analysis of data from an online panel survey. The results suggest that green consumer behaviors are positively related to subjectively evaluated resources such as feelings of economic sufficiency and other, more "relational" resources, including social networks and emotional support. Performing such behaviors may therefore lead to psychological gains. These findings do paint a rather positive picture of environmental behaviors, since they may thus be viewed as having a personal positive trade-off. Although directional effects cannot be firmly established, our study suggests that pro-environmental behavior may increase wellbeing and experienced prosperity. Future studies should further investigate these causalities and implications of these suggested relationships.

**Keywords:** subjective resources; scarcity; abundance; sufficiency; pro-environmental behavior; psychological gains; trade-offs; SAS framework; subjective well-being

## 1. Introduction

Resources are necessary prerequisites for all living organisms, humans among them, since they provide means to get needs and desires fulfilled—e.g., food, shelter, and social positions. However, the human condition is specific in that humans have the possibility to reflect upon resources, and these reflections have been the source of resource categorizations (based on classical economics, see e.g., [1,2]; standard ecology, see e.g., [3]; sociology, see e.g., [4]; or psychology, see e.g., [5]). Although resources come in many forms—they can be natural such as forests, economic such as money or capital, or psychological attributes such as charisma—it is hard to define exactly what is meant by the term "resources". However, a common view is that resources are what individuals sharing a similar cultural tradition see as intrinsically valued entities or as entities that can be traded, i.e., used to attain other valuable ends [4–7]. Further, people may find their resources scarce, sufficient, or abundant. In the emerging SAS framework, which we build on in this research, scarcity, sufficiency, and abundance are not objective concepts; there are always subjective components to these perceptions [4,8]. For instance,

scarce resources need to be perceived and evaluated as such in order for people to take action against further deterioration [9].

In the present research, we explore the relation between green consumer behaviors and subjectively perceived personal resources. In doing so, we ask what kind of gains and losses people experience when acting pro-environmentally. While we are dealing with a circular cause–effect relationship, where behaviors and subjective resources are likely to influence each other, we have here chosen to focus on the behavioral roots of resource perceptions, treating pro-environmental behaviors as predictive variables and subjective resources as dependent variables. The purpose of the study is therefore to contribute to the growing interdisciplinary research field of the psychology of sustainability.

## 1.1. The Psychology of Scarcity, Abundance, and Sufficiency (SAS)

History is full of examples where humans and societies threaten, exhaust, react to changes in, or preserve the resources they depend on [10,11], one of the most well-known being Thomas Hobbes' illustration of a "war of all against all" [12]. According to mainstream political theory, competition results from a scarcity of resources, while cooperation can be found in situations of resource abundance [13–16]. Importantly, however, resource perceptions can be modified by institutional arrangements, which help preserve resources [11,15,17,18] and prevent scarcity perceptions from leading to conflict.

While a lack of necessities for survival such as food, clean water, and shelter could be referred to as absolute scarcities, many of today's scarcities are relative in the sense that they instead concern situations when desires surpass the resources available [8]. For example, one may desire material resources, such as a new car or a bigger house, but also more time to develop personal interests, a larger social network, and more emotional support. In this vein, the SAS framework moves beyond defining and measuring such resources objectively, recognizing that resources become subjective when perceived and evaluated by an individual. An objective resource, such as "income" in terms of money, can then be separated from the individual perception of its value. Such subjective assessments are made in relation to locally perceived individual expenses, needs, or desires, and depend on a psychological process where comparisons are made between a perceived and a desired level of assets (see [4,6,9] for a discussion on the importance of comparisons).

Scholars from various disciplines have explored the links between behavior and the scarcity of personal resources. For example, some studies even suggest that confronting people with simple cues of scarcity is enough to make them act more selfishly [19,20] and pace a higher value on things they perceive as lacking than they usually do [21], which is pretty much in line with Hobbes' proposition [12]. Further, it is intuitive that an objective scarcity of material resources can cause psychological reactions such as stress and negative feelings. Research also shows that such scarcity may decrease attention and favor habitual behaviors, leading to irrational and short-sighted decision making [22]. Experienced scarcity of time is another example of how short-term perceived gains may have detrimental long-term effects. For instance, when approaching a deadline, objective as well as subjectively experienced time scarcity may increase focus and the ability to get immediate things done. However, such increased focus (labeled "tunneling") may come at the cost of unwanted side-effects caused by neglecting other important tasks such as paying the bills. Eventually, tunneling behavior risks causing collapse rather than productivity [23]. Clearly, objective resource scarcity (observed and measured) as well as subjective resource scarcity (perceived, assessed, and evaluated) can affect people's mental states and behaviors.

Previous research on resource assessments has focused mostly on one resource at a time [5] and has thereby ignored the core insight of the influential conservation of resources theory [24], which holds that resources can buffer as well as increase psychological stress [7]. Following this, from an SAS perspective, different resources are assumed to be interrelated. Perhaps the most striking consequence of experiencing personal resource scarcity is that it brings resource trade-offs to the top of the mind [19,25,26]. Experiencing scarcity of one type of resource, e.g., money, then leads individuals to think of trade-offs between money scarcity and other kinds of resources, including

emotional pay-offs. This implies that people may make trade-offs not only between objective resources, for example by paying for necessary groceries, but also between objective and subjective resources, for example paying extra for environmentally friendly groceries in order to feel good. The costs of pro-environmental consumer choices may then be "traded" against other, "softer" kinds of resources, possibly increasing the perceived value of social and/or emotional resources. Your willingness to spend effort, time, and money on something may in other words both affect, and be affected by, your subjective evaluation of such resources. Thus, the causality may here work in both directions. In this vein, the SAS framework can be used to identify and analyze both the causes and the effects of pro-environmental behavior.

Several previous studies show a positive relationship between pro-environmental behavior and subjective well-being [27–31]. Pro-environmental behavior has also been shown to correlate positively with so-called "constructive pessimism", i.e., with a pessimistic anticipation of future subjective well-being [32]. Future negative well-being (here understood as perceived scarcity) can then be avoided by performing "positive"/"constructive" pro-environmental behaviors today. Furthermore, significant correlations have been found between subjective perceptions of economic and socioemotional resources and well-being in general: the more one sees these resources as sufficient/abundant (scarce), the more positive (negative) one becomes about the future, the more (less) satisfied one becomes with life, and the less (more) worried one feels [6,9]. Other studies have shown that social resources, such as emotional support, are positively related to use of effective strategies for coping with stress [33,34], which can reduce undesirable stress-induced psychological symptoms [35], which may in turn affect sustainable behavior.

These findings all point to a positive relationship between subjectively assessed resources and well-being in terms of appraisals of life satisfaction and future expectations. Since pro-environmental behavior is also linked to well-being, which in turn depends on subjectively assessed resources, it can be assumed that such resources can be affected by pro-environmental behavior.

## 1.2. The Present Study

Researchers commonly seek to explain pro-environmental behavior by looking at individual differences in psychological variables (such as subjective resource assessments) in conjunction with socioeconomic background variables. However, in the consumer contexts investigated here, the potential behavioral effects of resource assessments are likely to be weak and confounded by more general explanatory variables such as income, gender, subjective well-being, trust, and social value orientation. Taking a different path of enquiry from previous studies, we instead believe that it would be fruitful to reverse the perspective and investigate resource trade-offs by looking at the unique effects that pro-environmental behaviors may have on subjectively assessed resources. We assume that consumers can experience gains or losses in terms of subjectively perceived personal resources as a result of behaving pro-environmentally.

Thus, this study investigates what kind of trade-offs people experience when they spend money and effort on environmentally friendly alternatives. Further, it looks at the nature of the gains and losses people experience when they act pro-environmentally. Specifically, we explore how various green consumer choices relate to subjective, individually assessed perceptions of different types of resources. We test the relationships between pro-environmental consumer behaviors (purchase of eco-labeled goods, green eating habits, and choice of transportation mode) and perceptions of economic, time, emotional, and social resources while controlling for other potentially confounding variables such as subjective well-being, social value orientation, trust, and background variables in hierarchical regression models. However, before proceeding with this analysis, we investigate the relationship between such green behaviors and subjective well-being. If this relationship can be confirmed, as would be expected in light of previous studies, it is motivated to further investigate whether pro-environmental behaviors have other, unique relationships with subjective resources. Statistically, these latter relationships are expected to be rather weak. Since our approach is novel

and our data are correlational, we recognize the difficulties implied in drawing strong conclusions about causality.

## 2. Materials and Methods

The investigation was conducted through an online survey sent to a random sample from a self-selected panel of respondents administered by the Department of Psychology, University of Gothenburg. The panel, which still exists, consists of a combination of community and students, and participants have had the opportunity to sign-in via advertisements on town property and at the university. The survey was slightly revised after a questionnaire had been pretested on a smaller sample of respondents. The final version of the survey was sent out in April 2017 to a random sample of 2892 individuals from the online panel, of whom 986 took the survey. Potential respondents were informed that there would be a prize draw of two gift cards worth 500 SEK each among those who completed the survey. After cleansing the dataset from respondents that did not finish the survey (the majority of these quit the survey after a few initial questions), $N = 880$ remained. All participants were informed about the general purpose of the study, along with contact information to the researchers. Further, respondents were informed that participation was voluntary, that their answers were anonymous, and that they could withdraw at any time. The analysis was conducted in SPSS using hierarchical regression analysis.

### 2.1. Subjective Resources

The most relevant types of subjective resources were measured through the use of the Relative Resource Assessment Scale in accordance with previous studies [6,9]. This recently developed scale was designed to measure subjective assessments of personal resources such as money, time, social networks, and emotional support. Respondents were asked to evaluate each type of resource in terms of scarcity, sufficiency, or abundance using different reference points, in three items: compared to my needs; compared to what I am used to; compared to other important persons in my life. Respondents answered on a 7-point scale ranging from −3 (indicating perceived scarcity) via 0 (indicating perceived sufficiency) to +3 (indicating perceived abundance). The scale was produced on the assumption that respondents would intuitively see a correspondence between sufficiency and zero on the scale, while abundance is reflected in positive values and scarcity is reflected in negative values.

### 2.2. Pro-Environmental Behavior

We are primarily interested in regressing pro-environmental behaviors on subjective resources. The idea was to cover the most important aspects of individual everyday behaviors with potential environmental impact, thus excluding less frequent behaviors, such as vacation traveling, buying a car, choosing a home, etc. We therefore chose to investigate grocery shopping, the meat/fish consumption, and everyday travel modes. Furthermore, we distinguished between recent choices (e.g., what one ate yesterday) and typical/habitual behavior (e.g., how often one eats each foodstuff).

Grocery shopping last time was measured by asking respondents to think about the last time they bought milk/dairy products, fruits/vegetables, eggs, flour/grain/cereals/muesli, meat/fish, canned food, oil, and coffee/tea, and indicate whether they bought an eco-labeled product or not. Respondents could answer that they never buy the product in question. Buying non-eco-labeled (i.e., conventional) products equaled 1, and buying eco-labeled products equaled 2, indicating that a higher value corresponds to a more environmentally friendly the behavior. It could be debated whether the third option (3), never buying a product (which is a necessary option for respondents), is part of the same scale. For instance, never buying eggs (because you never eat eggs) could in one way be viewed as the most pro-environmental behavior (it implies less environmental impact than buying any kind of eggs), but taking this into account would be to mix different logics: pro-environmental behavior in terms of buying eco-labeled products and pro-environmental behavior more generally. Only the former category was of interest for this study. Therefore, answers of 3 on these items were recoded as

the mean value of the responses on the scale 1–2. The eight items were then averaged into an index variable. Hence, an interval scale ranging from 1 to 2 was created (Cronbach's alpha 0.756).

Grocery shopping habits were measured by asking respondents how often they buy environmentally labeled milk/dairy products, fruits/vegetables, eggs, flour/grain/cereals/muesli, meat/fish, canned food, oil, and coffee/tea. Answers could range from (1) "I never buy environmentally labeled products from this category" to (7) "I always buy environmentally labeled products from this category", with the middle option (4) being "About half of all my purchases (from this category) are environmentally labeled". There was also the possibility (8) of indicating that "I never buy products from this category", which was recoded as the mean value of the responses on the scale 1–7 (similar to how we processed this answer for the variable grocery shopping last time). The eight items were averaged into an index variable ranging from 1 to 7 (Cronbach's alpha 0.879).

The variable ate yesterday was intended to measure the consumption of meat and fish, which on a general level is known to have negative environmental impacts compared to vegetables and legumes. Respondents were asked if they ate any of eight types of meat, fish, or shellfish, and if the product was eco-labeled. Not having eaten the product at all scored the highest value (3), and eating a non-eco-labeled product scored the lowest (1), while eating an environmentally labeled product scored 2. We tested this variable with both weighted and unweighted meat/fish items. In the weighted analysis, each item was weighted according to its estimated environmental impact by the National Food Agency [36]. How to calculate this is not self-evident, given that what is bad from a climate change perspective is not necessarily bad from a biological diversity or an eutrophication perspective. We chose to focus on emissions of greenhouse gases, since climate considerations more likely guided participants' choices than for example biological diversity or eutrophication. This implied that beef scored 10 times higher in environmental impact than poultry, fish and shellfish did, four times higher than pork and game did, and 1.5 times higher than lamb did. The eight items were indexed into a single variable ranging from 1 to 3 (Cronbach's alpha 0.625).

Eating habits were measured using the same eight items as in ate yesterday, and we tested both the unweighted and the weighted variable. Here, respondents were asked to indicate how often they eat each foodstuff—ranging from (1) "Every day" to (7) "I never eat food from this category". The eight weighted items were indexed into a single variable ranging from 1 to 7 (Cronbach's alpha 0.855).

Turning to transportation issues, travel mode last week was measured by asking respondents by which mode of transportation they mostly employed to travel to work/school/corresponding daily activity during the last week. Possible answers were (1) by car, by myself, (2) by car, with others, (3) by public transportation (bus, tram, railway etc.), (4) by bicycle, and (5) by foot, indicating that low scores were the least environmentally friendly, and high scores were considered the most environmentally friendly.

Travel mode habits were measured by asking respondents to indicate on a scale ranging from (1) "every day" to (6) "never" how often they travel to work/school/corresponding daily activity by the same five travel modes described in the *travel mode last week* variable. We averaged these items into a single variable, testing variants with all five items, leaving out "by car, with others" and leaving out "by car, with others" and "by car, by myself", but all indices scored low on Cronbach's alpha (see Table 1), and this variable was hence not included in the analyses.

## 2.3. Subjective Well-Being

Subjective well-being was measured using the established Satisfaction With Life Scale [37], where respondents answer five items ("In most ways, my life is close to my ideal"; "The conditions of my life are excellent"; "I am satisfied with my life"; "So far, I have gotten the important things I want in life"; "If I could live my life over, I would change almost nothing") on a 7-point Likert scale ranging from strongly disagree (1) to strongly agree (7), which were then indexed to a single variable (Cronbach's alpha 0.872).

**Table 1.** Variables included in the analyses, mean values, standard deviations, scales, *N* values, and Cronbach's alpha.

| Variables | Mean | Std. dev. | Scale | N | Cronbach's Alpha |
|---|---|---|---|---|---|
| Subjective resources: economic | −0.30 | 1.49 | −3 to +3 | 880 | 0.861 |
| Subjective resources: time | −0.47 | 1.47 | −3 to +3 | 880 | 0.868 |
| Subjective resources: social networks | −0.15 | 1.30 | −3 to +3 | 880 | 0.841 |
| Subjective resources: emotional support | 0.31 | 1.43 | −3 to +3 | 877 | 0.881 |
| Grocery shopping last time | 1.59 | 0.27 | 1–2 | 876 | 0.756 |
| Grocery shopping habits | 3.93 | 1.36 | 1–7 | 876 | 0.879 |
| Ate yesterday weighted | 2.74 | 0.37 | 1–3 | 874 | 0.625 |
| Ate yesterday unweighted | 2.75 | 0.33 | 1–3 | 874 | 0.625 |
| Eating habits weighted | 5.20 | 1.23 | 1–7 | 878 | 0.855 |
| Eating habits unweighted | 5.16 | 1.11 | 1–7 | 878 | 0.855 |
| Travel mode last week | 3.24 | 1.00 | 1–5 | 878 | N.A. |
| Travel mode habits (by car by myself, by car with others, public transportation, bicycle, walking) | 4.15 | 0.70 | 1–6 | 880 | 0.114 |
| Travel mode habits (by car by myself, public transportation, bicycle, walking) | 3.85 | 0.80 | 1–6 | 880 | 0.030 |
| Travel mode habits (by public transportation, bicycle, walking) | 3.43 | 0.85 | 1–6 | 880 | 0.374 |
| Subjective well-being | 4.64 | 1.33 | 1–7 | 879 | 0.872 |
| Social value orientation | 33.30 | 12.84 | −14.26 to +89.61 (empirically) | 878 | N.A. |
| Generalized trust | 6.27 | 2.33 | 0–10 | 880 | N.A. |
| Trust in environmental institutions | 3.74 | 0.73 | 1–5 | 879 | 0.883 |
| Age | 28.64 | 9.36 | 18–78 | 877 | N.A. |
| Gender | 1.20 | 0.40 | Female–Male (1–2) | 854 | N.A. |
| Income | 2.52 | 1.97 | 1–12 (5000 SEK intervals. starting at ≤10,000 SEK and ending at ≥60,000 SEK/month) | 877 | N.A. |

*2.4. Trust*

Trust is assumed to be a measure of social capital [38–40], and thus important for the level of cooperation in a society. The more people find fellow citizens trustworthy, the more likely it is that they will have positive attitudes to contributing to the collective good. It is quite intuitive that a person who trusts anonymous others will also be positive in her evaluation of resources in terms of SAS. Therefore, in our models, we control for generalized trust as a social resource that may influence subjective evaluations of other resources, as well as subjective well-being.

Generalized trust was measured by asking: "In your opinion, to what extent can people in general be trusted?" Respondents were requested to indicate their answer on an 11-point scale ranging from 0 ("You cannot trust people in general") to 10 ("You can trust people in general").

Trust in political institutions is related to generalized trust, and studies show that a positive evaluation of the trustworthiness and legitimacy of institutions plays an important role for the acceptance of and compliance with government rules [41–43]. Evidence is strong for the impact of generalized trust in political institutions on environmental policy attitudes [44–50]. However, it has also been argued that trust in specific institutions is more important than general trust in political institutions, i.e., that taking a specific institution into account is more important for the understanding and acceptance of specific policy measures [43,51] Given this, we find it significant to include trust in environmental institutions as a control variable, since it could correlate with both the behaviors and the subjective resources we investigate.

Trust in environmental institutions was measured by asking respondents to what extent they trust that different authorities and other actors provide correct and objective information about the

environmental consequences of people's everyday behavior and consumption. The actors included were the Ministry of Environment and Energy, Ministry of Finance, Swedish Environmental Protection Agency, Swedish Chemicals Agency, Swedish Agency for Marine and Water Management, Swedish Society for Nature Conservation, and environmental scientists. Answers could range from very low trust (1) to very high trust (5). Answers on these items were indexed into a single variable (where the Ministry of Finance was omitted, since it lowered the Cronbach's alpha score; the Ministry of Finance was included in the questionnaire since this institution actually influences environmental policy. However, it is unlikely that respondents are actively aware of this fact) (Cronbach's alpha 0.883).

### 2.5. Social Value Orientation

For nearly 50 years, researchers have tried to capture people's motives in social dilemma situations through the concept of social value orientation (SVO), showing how people evaluate the resource outcomes for themselves and others [52]. A vast literature has investigated the importance of individuals' fundamental value orientations in this context, concluding that there are three core social value orientations that differentiate people: people who are prosocial, individualists, and those who are competitive [53,54]. Others have added altruists as a variation of prosocials in a fourth category [55]. Prosocial individuals (especially altruists) are most inclined to cooperate in social dilemma situations, while individualists and competitive persons have variations of the pro-selfish characteristic, and thus do not choose cooperative behaviors to the same extent [56,57]. As a consequence, social value orientation is included as a control variable in the analyses, since this factor could potentially correlate with subjective well-being, the pro-environmental behaviors, and the subjective resources we investigate.

Social value orientation was measured using a slider measure where respondents were asked to allocate money to themselves and another anonymous person. Depending on how much was given to themselves and the other person over six items, each respondent was given a score ranging from altruism to competitiveness, with prosocial and individualism in between (see [55] for how to calculate scores).

### 2.6. Background Variables

Income was measured by requesting respondents to indicate their monthly gross income in one of 12 categories, ranging from less than 10,000 SEK to more than 60,000 SEK in 5000-SEK intervals. Gender could be answered as female (1), male (2), or other/do not want to indicate (3; excluded from the analysis). Age was measured by asking respondents to indicate their age in numerical values.

## 3. Results

Given the results from previous studies, it is likely that subjective resources correlate with subjective well-being [6,9]. Hence, we first regressed the behavioral variables on subjective well-being, controlling for generalized trust, social value orientation, and background variables. Since previous research suggests that pro-environmental behaviors also correlate with well-being [27–31], multicollinearity between subjective well-being and pro-environmental behaviors may be a potential problem. Although our analyses indicate the expected correlations between pro-environmental behaviors and subjective well-being, multicollinearity did not prove to be problematic, indicated by VIF (Variance Inflation Factor) values in Table 2, ranging between 1.06–2.34, i.e., considerably below levels usually considered problematic [58]. As shown in Table 2, both grocery shopping last time (positive correlation) and eating habits (negative correlation) predict subjective well-being. As expected, generalized trust and background variables (age, gender, and income) also influence subjective well-being. The pattern is similar when entering the unweighted eating variables instead of the weighted (see Table S1). This implies that the data quality is good (results are in line with previous studies and theoretical expectations), and that it is plausible to move onto the main aim of the study, which was to investigate the relationship between pro-environmental behaviors and subjective resources.

**Table 2.** Hierarchical multiple regression model testing the relation between pro-environmental behaviors and subjective well-being. Unstandardized correlations *(B)*, standard error *(SE)*, and collinearity diagnostics *(VIF)*. N = 835.

| Variables | Model [1] | | | | | | |
|---|---|---|---|---|---|---|---|
| | 1 | | 2 | | 3 | | |
| | *B* | *SE* | *B* | *SE* | *B* | *SE* | *VIF* |
| Grocery shopping last time | 0.660 ** | 0.253 | 0.669 ** | 0.243 | 0.603 * | 0.241 | 2.244 |
| Grocery shopping habits | 0.009 | 0.050 | −0.046 | 0.049 | −0.038 | 0.049 | 2.341 |
| Ate yesterday (weighted) | 0.0042 | 0.131 | −0.028 | 0.127 | −0.056 | 0.126 | 1.153 |
| Eating habits (weighted) | −0.054 | 0.040 | −0.063 | 0.039 | −0.086 * | 0.039 | 1.222 |
| Travel mode last week | 0.055 | 0.046 | 0.046 | 0.044 | 0.055 | 0.045 | 1.088 |
| Generalized trust | | | 0.160 *** | 0.004 | 0.163 *** | 0.020 | 1.082 |
| Social value orientation | | | 0.003 | 0.020 | 0.003 | 0.004 | 1.086 |
| Age | | | | | −0.019 ** | 0.006 | 1.515 |
| Gender [2] | | | | | −0.308 ** | 0.110 | 1.059 |
| Income | | | | | 0.078 ** | 0.027 | 1.526 |
| Adj. $R^2$ | 0.015 ** | | 0.091 *** | | 0.112 *** | | |
| $\Delta R^2$ | 0.015 ** | | 0.076 *** | | 0.021 *** | | |
| F | 3.626 ** | | 12.991 *** | | 11.482 *** | | |

[1] Subjective well-being modeled as dependent variable. [2] Male is numerically higher. *** $p < 0.001$, ** $p < 0.01$, * $p < 0.05$.

The regression model of subjective evaluation of economic resources (Table 3) can explain a substantial part of the variance (*Adj. $R^2$* = 0.261). Eating habits have a significant positive correlation with subjective evaluation of economic resources, which implies that the greener a person's diet is in terms of how often s/he eats meat/fish, generating emissions of greenhouse gases, the more money s/he thinks s/he has. Thus, green eating habits seem to relate to feeling more economic abundance. Travel mode last week also correlates with the dependent variable, implying that the more environmentally friendly one traveled, the higher the evaluation of economic resources. Further, there is a robust positive correlation between subjective well-being and evaluation of economic resources, which is in line with previous research. Also, as expected, objective income and subjective evaluation of economic resources correlate significantly. Younger persons are also more satisfied with their economic resources than older persons are. Finally, generalized trust and social value orientation correlate with a subjective evaluation of economic resources: the more you trust people in general and the more prosocial you are, the more satisfied you are with the economic resources you have. Suspected multicollinearity problems proved to be unfounded (this is true for all analyses of subjective resources as dependent variables, see Tables 3–6). Since it may be argued that individuals are not aware that different types of meat/fish have different environmental impacts, we tested entering the unweighted eating variables instead of the weighted (see Table S2). This did not change the patterns displayed in Table 3.

**Table 3.** Hierarchical multiple regression model testing the relation between pro-environmental behaviors and subjective economic resources. Unstandardized correlations *(B)*, standard error *(SE)*, and collinearity diagnostics *(VIF)*. N = 834.

| Variables | Model [1] | | | | | | |
|---|---|---|---|---|---|---|---|
| | 1 | | 2 | | 3 | | |
| | *B* | *SE* | *B* | *SE* | *B* | *SE* | *VIF* |
| Grocery shopping last time | 0.600 * | 0.287 | 0.383 | 0.272 | 0.250 | 0.249 | 2.258 |
| Grocery shopping habits | −0.052 | 0.057 | −0.079 | 0.055 | −0.065 | 0.050 | 2.351 |
| Ate yesterday (weighted) | −0.121 | 0.149 | −0.175 | 0.141 | −0.190 | 0.130 | 1.155 |
| Eating habits (weighted) | 0.084 † | 0.045 | 0.096 * | 0.043 | 0.121 ** | 0.040 | 1.230 |
| Travel mode last week | 0.029 | 0.052 | 0.015 | 0.050 | 0.101 * | 0.047 | 1.103 |
| Subjective well-being | | | 0.328 *** | 0.039 | 0.285 ** | 0.036 | 1.145 |



**Table 3.** *Cont.*

| Variables | Model [1] | | | | | | |
|---|---|---|---|---|---|---|---|
| | 1 | | 2 | | 3 | | |
| | *B* | *SE* | *B* | *SE* | *B* | *SE* | *VIF* |
| Social value orientation | | | 0.006 | 0.004 | 0.009 * | 0.004 | 1.088 |
| Generalized trust | | | 0.073 ** | 0.024 | 0.070 ** | 0.022 | 1.259 |
| Trust in environmental institutions | | | −0.090 | 0.072 | −0.084 | 0.066 | 1.158 |
| Age | | | | | −0.045 *** | 0.006 | 1.536 |
| Gender [2] | | | | | 0.220 † | 0.114 | 1.067 |
| Income | | | | | 0.353 *** | 0.028 | 1.541 |
| Adj. $R^2$ | 0.006 † | | 0.117 *** | | 0.261 *** | | |
| $\Delta R^2$ | 0.006 † | | 0.111 *** | | 0.144 *** | | |
| F | 2.032 † | | 13.270 *** | | 25.459 *** | | |

[1] Subjective economic resources modeled as dependent variable. [2] Male is numerically higher. *** $p < 0.001$, ** $p < 0.01$, * $p < 0.05$, † $p < 0.10$.

Turning to the subjective resource of time as a dependent variable (Table 4; cf. Table S3), there is only a small tendency ($p < 0.10$) that the more environmentally friendly a person's travel mode was over the last week, the more likely it is that s/he finds their time resources abundant. No other behavioral variable correlates significantly with the dependent variable. The few significant correlations in this model reflects its low explained variance (*Adj. $R^2$* = 0.014).

**Table 4.** Hierarchical multiple regression model testing the relation between pro-environmental behaviors and the subjective resource time. Unstandardized correlations *(B)*, standard error *(SE)*, and collinearity diagnostics *(VIF)*. N = 834.

| Variables | Model [1] | | | | | | |
|---|---|---|---|---|---|---|---|
| | 1 | | 2 | | 3 | | |
| | *B* | *SE* | *B* | *SE* | *B* | *SE* | *VIF* |
| Grocery shopping last time | −0.163 | 0.283 | −0.221 | 0.284 | −0.193 | 0.284 | 2.258 |
| Grocery shopping habits | −0.001 | 0.056 | −0.010 | 0.057 | −0.005 | 0.057 | 2.351 |
| Ate yesterday (weighted) | 0.073 | 0.147 | 0.054 | 0.147 | 0.037 | 0.148 | 1.155 |
| Eating habits (weighted) | 0.017 | 0.045 | 0.018 | 0.045 | 0.024 | 0.046 | 1.230 |
| Travel mode last week | 0.119 * | 0.052 | 0.123 * | 0.052 | 0.099 † | 0.053 | 1.103 |
| Subjective well-being | | | 0.080 * | 0.040 | 0.091 * | 0.041 | 1.145 |
| Social value orientation | | | 0.004 | 0.004 | 0.004 | 0.004 | 1.088 |
| Generalized trust | | | 0.039 | 0.025 | 0.038 | 0.025 | 1.259 |
| Trust in environmental institutions | | | −0.107 | 0.075 | −0.112 | 0.075 | 1.158 |
| Age | | | | | 0.002 | 0.007 | 1.536 |
| Gender [2] | | | | | 0.170 | 0.130 | 1.067 |
| Income | | | | | −0.058 † | 0.032 | 1.541 |
| Adj. $R^2$ | 0.003 | | 0.011 * | | 0.014 | | |
| $\Delta R^2$ | 0.003 | | 0.008 * | | 0.003 | | |
| F | 1.440 | | 1.990 * | | 1.979 * | | |

[1] Subjective resource time modeled as dependent variable. [2] Male is numerically higher. *** $p < 0.001$, ** $p < 0.01$, * $p < 0.05$, † $p < 0.10$.

Table 5 (cf. Table S4) displays the analysis of the resource social networks as a dependent variable. Grocery shopping habits come out as having a positively significant correlation with social networks. The more environmentally labeled products a person buys, the more social networks she feels s/he has. Further, there is an expected positive relationship between subjective well-being and social networks, and the results also indicate that men valuate their social networks higher than women. The explanatory power of the regression model is moderate (*Adj. $R^2$* = 0.139).

**Table 5.** Hierarchical multiple regression model testing the relation between pro-environmental behaviors and the subjective resource social networks. Unstandardized correlations *(B)*, standard error *(SE)*, and collinearity diagnostics *(VIF)*. N = 834.

| Variables | Model [1] | | | | | | |
|---|---|---|---|---|---|---|---|
| | 1 | | 2 | | 3 | | |
| | B | SE | B | SE | B | SE | VIF |
| Grocery shopping last time | −0.105 | 0.250 | −0.329 | 0.234 | −0.336 | 0.234 | 2.258 |
| Grocery shopping habits | 0.106 * | 0.050 | 0.090 [†] | 0.047 | 0.100 * | 0.047 | 2.351 |
| Ate yesterday (weighted) | 0.150 | 0.129 | 0.111 | 0.121 | 0.087 | 0.122 | 1.155 |
| Eating habits (weighted) | −0.043 | 0.040 | −0.029 | 0.037 | −0.017 | 0.038 | 1.230 |
| Travel mode last week | 0.021 | 0.046 | 0.007 | 0.043 | 0.005 | 0.044 | 1.103 |
| Subjective well-being | | | 0.336 *** | 0.033 | 0.335 *** | 0.034 | 1.145 |
| Social value orientation | | | 0.006 [†] | 0.003 | 0.006 [†] | 0.003 | 1.088 |
| Generalized trust | | | 0.036 [†] | 0.020 | 0.035 [†] | 0.020 | 1.259 |
| Trust in environmental institutions | | | −0.076 | 0.062 | −0.079 | 0.062 | 1.158 |
| Age | | | | | −0.010 [†] | 0.006 | 1.536 |
| Gender [2] | | | | | 0.220 * | 0.107 | 1.067 |
| Income | | | | | 0.035 | 0.026 | 1.541 |
| Adj. $R^2$ | 0.005 [†] | | 0.134 *** | | 0.139 [†] | | |
| $\Delta R^2$ | 0.005 [†] | | 0.129 *** | | 0.005 [†] | | |
| F | 1.880 [†] | | 15.364 *** | | 12.215 *** | | |

[1] Subjective resource social networks modeled as dependent variable. [2] Male is numerically higher. *** $p < 0.001$, ** $p < 0.01$, * $p < 0.05$, [†] $p < 0.10$.

A similar pattern appears in the analysis of emotional support as a dependent variable (Table 6; cf. Table S5). Here too, grocery shopping habits are positively correlated with the dependent variable, but this is only significant at the $p > 0.10$ level. There is also further support for the positive relationship between subjective well-being and subjective resources (here: emotional support). Generalized trust is clearly related with emotional support as well. The explained variance for this regression model is moderate to good (*Adj. $R^2$* = 0.199).

**Table 6.** Hierarchical multiple regression model testing the relation between pro-environmental behaviors and the subjective resource emotional support. Unstandardized correlations *(B)*, standard error *(SE)*, and collinearity diagnostics *(VIF)*. N = 833.

| Variables | Model [1] | | | | | | |
|---|---|---|---|---|---|---|---|
| | 1 | | 2 | | 3 | | |
| | B | SE | B | SE | B | SE | VIF |
| Grocery shopping last time | 0.087 | 0.273 | −0.195 | 0.248 | −0.194 | 0.248 | 2.258 |
| Grocery shopping habits | 0.101 [†] | 0.054 | 0.081 | 0.050 | 0.087 [†] | 0.050 | 2.351 |
| Ate yesterday (weighted) | 0.161 | 0.142 | 0.119 | 0.129 | 0.110 | 0.130 | 1.155 |
| Eating habits (weighted) | 0.021 | 0.043 | 0.043 | 0.039 | 0.055 | 0.040 | 1.230 |
| Travel mode last week | 0.075 | 0.050 | 0.051 | 0.045 | 0.050 | 0.047 | 1.103 |
| Subjective well-being | | | 0.440 *** | 0.035 | 0.443 *** | 0.036 | 1.145 |
| Social value orientation | | | 0.002 | 0.004 | 0.003 | 0.004 | 1.090 |
| Generalized trust | | | 0.053 * | 0.022 | 0.052 * | 0.022 | 1.262 |
| Trust in environmental institutions | | | −0.061 | 0.065 | −0.064 | 0.065 | 1.159 |
| Age | | | | | −0.004 | 0.006 | 1.535 |
| Gender [2] | | | | | 0.199 [†] | 0.114 | 1.067 |
| Income | | | | | 0.017 | 0.028 | 1.541 |
| Adj. $R^2$ | 0.014 ** | | 0.198 *** | | 0.199 | | |
| $\Delta R^2$ | 0.014 ** | | 0.184 *** | | 0.001 | | |
| F | 3.339 ** | | 23.841 *** | | 18.186 *** | | |

[1] Subjective resource emotional support modeled as dependent variable. [2] Male is numerically higher. *** $p < 0.001$, ** $p < 0.01$, * $p < 0.05$, [†] $p < 0.10$.

## 4. Discussion

The results reveal several interesting patterns when it comes to relations between the measured behaviors and the subjective resource evaluations. Eating habits relate positively to the perception of economic resources, meaning that the more pro-environmental eating habits a person has, the more affluent s/he feels in economic terms. The same can be said about travel modes during the last week; the more environmentally friendly the travel mode, the more people tend to assess their economic resources as abundant. Further, buying eco-labeled groceries is positively related to the relational resources of social networks and emotional support; thus, pro-environmental choices in the grocery store relate to perceived abundance of social networks and emotional support. As expected, subjective well-being also had positive relationships with all the subjective resources measured here, further strengthening this relationship found in previous research [6,9]. Another interesting finding is that generalized trust correlates positively with the measures of subjective economic resources and emotional support and weakly also with social networks; the more you trust your fellow citizens, the more likely it is that you are satisfied with your economic resources and emotional support. This is in line with established knowledge of the importance of trust for people's life satisfaction [59].

Our interpretation of these results is that performing pro-environmental behaviors is positively related to psychological gains. The expected [27–31] and observed relationship between recent purchases of eco-labeled groceries and subjective well-being strengthens this interpretation further. Although it could be expected that buying an eco-labeled product would be viewed as a sacrifice that could affect the evaluation of economic resources negatively, we find no support for such expectations. Further, scoring high on environmental grocery shopping habits seems to be positively correlated also to other, more "relational" resources. This finding is in line with previous research, that suggests social inclusion, i.e., experiencing a surplus of human relations, as a factor that increases the likelihood of prosocial behavior [60].

We also found a weak negative correlation between eating habits and subjective well-being (see Table 2), indicating that the more environmental impact from people's eating choices, the higher their subjective well-being. This can be interpreted as a hedonistic expression of gaining satisfaction from "living the good life", in which meat consumption is highly valued—and also related to social status [61,62]. This result suggests that green behaviors do not automatically result in higher levels of well-being—and that subjective well-being and subjective resource evaluations are different things; as may be seen in Tables 3–6, no significant negative correlations were found between eating habits and subjective resources.

In contrast, the analysis of the subjective resource time does not demonstrate the same pattern. As may be seen in Table 4, there is a weak tendency that travel mode last week relates positively with the subjective assessment of the resource of time (significant only at the $p < 0.10$ level). We would have expected a stronger relationship, since environmentally friendly traveling modes such as traveling collectively, bicycling, or walking often are considered more time consuming than driving a car. People who feel pressed for time in their lives may not consider traveling collectively to the same extent as others. However, the explained variance in this regression model is low, which is reflected in that even subjective well-being is weakly correlated with the assessment of time resources. With the minor exception of income, no other variable were found to correlate with the time resource variable. Thus, these results should be interpreted with caution. It is possible that the subjective assessment of the resource of time is relatively unimportant in the choice situations we investigate here.

## 5. Conclusions and Implications

The general research interest in subjective resources is relatively new and promising. Understanding the importance of individuals' subjective resources adds to our knowledge of attitude formation and behaviors more generally. It also contributes to our understanding of the impact of objective resources on psychological aspects. To our knowledge, no previous studies have dealt with the relationship between pro-environmental behaviors and subjective assessments of resources. This study

not only demonstrates the existence of such a relationship, but it also shows that environmentally sustainable behaviors may have positive trade-offs in terms of psychological gains. This is a first contribution to an overlooked research field that needs to be investigated further, and which we hope to have demonstrated is theoretically interesting.

Our results do paint a rather positive picture of promoting environmental behavior. It seems that costly green behaviors may make people feel more prosperous, not only in terms of money, but also in terms of psychologically important resources such as social networks and emotional support. As discussed earlier, it is of course possible to interpret our results as springing from the impact of perceived resources on environmental behaviors, since our data do not admit a proper analysis of cause–effects. However, given the theoretical argument in this paper, we find it plausible to also see a reverse relationship, so that pro-environmental behaviors may lead to positive psychological trade-offs. Most likely, we are dealing with a circular cause–effect relationship. Future studies could further examine these possible psychological gains that may result from green behaviors. Further analysis of the policy implications is also warranted.

**Supplementary Materials:** The following are available online at http://www.mdpi.com/2071-1050/11/19/5467/s1, Table S1: Hierarchical multiple regression model testing the relation between pro-environmental behaviors and subjective well-being (with unweighted eating variables), Table S2: Hierarchical multiple regression model testing the relation between pro-environmental behaviors and subjective economic resources (with unweighted eating variables), Table S3: Hierarchical multiple regression model testing the relation between pro-environmental behaviors and the subjective resource time (with unweighted eating variables), Table S4: Hierarchical multiple regression model testing the relation between pro-environmental behaviors and the subjective resource social networks (with unweighted eating variables), Table S5: Hierarchical multiple regression model testing the relation between pro-environmental behaviors and the subjective resource emotional support (with unweighted eating variables).

**Author Contributions:** Conceptualization, M.Z., S.M., and L.-O.J.; methodology, M.Z. and L.-O.J.; software, M.Z.; validation, M.Z.; formal analysis, M.Z.; investigation, M.Z.; resources, L.-O.J.; data curation, M.Z.; writing—original draft preparation, M.Z.; writing—review and editing, M.Z., S.M., and L.-O.J.; visualization, M.Z.; supervision, M.Z.; project administration, M.Z.; funding acquisition, L.-O.J.

**Funding:** This research was funded by The Swedish Research Council (Vetenskapsrådet), grant number 421-2013-1732.

**Acknowledgments:** The authors would like to thank Isak Barbopoulos for assisting in collecting survey data and in transforming survey questions into the online software tool. The authors would also like to thank one anonymous reviewer for helpful suggestions, and Celia Aijmer Rydsjö for helping to improve the language.

**Conflicts of Interest:** The authors declare no conflict of interest. The funders had no role in the design of the study; in the collection, analyses, or interpretation of data; in the writing of the manuscript, or in the decision to publish the results.

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
