# Peer review of "On the Relationship between Pro-Environmental Behavior, Experienced Monetary Costs, and Psychological Gains"

_sustainability, doi:10.3390/su11195467_

Round 1

Reviewer 1 Report

This paper raises interesting questions and provides a novel way to look at environmental behavior. It raises the question of whether any random behavior also correlates to satisfaction -- but that would be another story.

The paper is a tad tedious to read and there are minor grammatical errors. If there is a way to shorten sentences and make them more direct and specific, that would help. Some terms need to be defined or explained, such as SAS framework and Matthew-rule. In the introduction, the lack of clarity comes from use of the term “resources” to refer to both natural resources and personal resources (e.g. time, wealth, social status). Because both resource types are relevant to the study, it is critical that you continue to specify the type of resources you are referring to.

Although the findings of this study have implications for natural resources, it is personal resources that are central to the design of the study; therefore, I would like to see discussion of natural resources largely replaced with relevant literature on personal resources and their relationship to decision making. This should include more information the perception of abundance of personal resources. Also, please make sure that the literature review content is directly relevant to your study and, where necessary, make the relevance explicit. For instance, I don’t immediately see how “tunneling” pertains to your study; help me see it.

The objectives of the study should be stated more clearly in the abstract and early in the introduction. It wasn’t until the end of the literature review that I understood that “In this study we are primarily interested in how various everyday consumer behaviors with potential environmental impacts affect people’s evaluation of costs and benefits in terms of the most important subjective resources (money, time, social networks, and emotional support).”

The abstract states “We find no support for the view that performing these behaviors should be understood as a sacrifice that affects negatively the subjective evaluation of resources.” I didn’t see where this was explicitly evaluated, but it seems like a valuable finding.

You state, “We are particularly interested in the constitution of subjective resources, which are therefore treated as dependent variables, while acknowledging that we might be dealing with a circular cause-effect relationship. In theory, environmental behavior may influence subjective evaluations of resources as well as the other way around. We do not, however, investigate nor argue in terms of causality in this paper.” It is not intuitive how subjective resources are dependent on PEBs, so this needs to be delved into and thoroughly explained. If you do not do so, your conclusions appear tenuous. For example, the conclusion that “the greener a person’s diet is…the more money she thinks she has,” is difficult to understand at this point. Temporally, the opposite seems appropriate – PEBs depend on
subjective resources.

I would like more information on the steps taken to maximize validity. For example, was the survey pretested and how was the self-selection sample conducted.

Some additional comments:

Line 45 -- threaten not threat?

Line 54 -- This statement sounds like subjectivity of resources is the independent variable and behavior is dependent. You later state it could well be a circular system -- which seems true.

Line 75 -- And this is the first statement of exploring the reverse relationship. But I'm not sure why this is important. More of an introduction to how behavior affects perception would be useful.

Line 87 -- resources' not resources?

Line 106 - types not type?

Line 367 - can partly predict?

Line 376 -- date are reliable

Reviewer 2 Report

This paper employed hierarchical regression analysis to investigate the relationship of pro-environmental behavior and the perception of subjective resources (economic, time, social network and emotional support). The results showed that pro-environmental behaviors are related to the perception of subjective resources and these behaviors seem to exhibit psychological gains for consumers.

The authors may consider to address the following issues:

1.      It seems that some of the variables may not be properly measured or the authors may consider to provide more supports on the measurements. For example, subjective resources, the major dependent variables, have a scale of -3 to 3. The means of the four individual items (economic, time, social network and emotional support) have a mean close to zero. This indicates that it is quite possible to have an unstable result of regression coefficients (positive or negative). Since these are measures of perception about the resources, why not just use a 1 to 7 scale? The other example is the measure of Gender and Grocery shopping last time. Both of them may be considered to use dummy variables instead of regular ones. From tables 2, we can see a negative regression coefficient for Gender. How to interpret the negative regression coefficient?

2.      It Table 2, subjective wellbeing is treated as dependent variable and we can find that it relates to several pro-environmental behaviors; and in Table 3, subjective wellbeing is treated as independent variable, together with several pro-environmental behaviors. It suggests there may exist potential multicollinearity problem among the independent variables.

3.      The authors may consider to explain more on the survey process of this on-line survey. For example, it seems that the questionnaire is quite complicated so it will be better for the authors to show the validity of the survey.  

Round 2

Reviewer 1 Report

My primary concern with this study is the research design, specifically the selection of dependent and independent variables. Although disclaimers are made that there may be a circular cause-and-effect relationship between PEBs and perceptions of subjective resources, the underlying assumption upon which the study is based is that the primary direction of the relationship between PEBs and subjective resources is that of PEBs affecting perceptions of subjective resources, not the other way around. This is not supported. In sections 1 and 1.1 some literature is cited to provide justification, but the examples not similar enough to the actual resources examined to support the research design. I suggest a thorough review of the literature to explore the reverse of the assumed relationship, subjective resources affecting PEBs. After reviewing the literature, present both sides and make the case that the primary direction of the relationship is reflected in the DVs and IVs you selected.

If the literature does not currently exist to allow you to determine the primary directional relationship between PEBs and perceptions of subjective resources, then I suggest that these results be presented from the beginning as only exploratory in nature. No causation should be implied, only relationships between variables should be presented. I fear that if the study is published as it is currently presented, that readers could make inappropriate conclusions. The exploratory nature of the study needs to be driven home.

In addition, the writing has been improved, though there are still improvements to be made in the structure and word choices. A writing professional may help improve clarity and flow.

Author Response

Comments by and responses to Reviewer 1

Comments:

"My primary concern with this study is the research design, specifically the selection of dependent and independent variables. Although disclaimers are made that there may be a circular cause-and-effect relationship between PEBs and perceptions of subjective resources, the underlying assumption upon which the study is based is that the primary direction of the relationship between PEBs and subjective resources is that of PEBs affecting perceptions of subjective resources, not the other way around. This is not supported. In sections 1 and 1.1 some literature is cited to provide justification, but the examples not similar enough to the actual resources examined to support the research design. I suggest a thorough review of the literature to explore the reverse of the assumed relationship, subjective resources affecting PEBs. After reviewing the literature, present both sides and make the case that the primary direction of the relationship is reflected in the DVs and IVs you selected.

If the literature does not currently exist to allow you to determine the primary directional relationship between PEBs and perceptions of subjective resources, then I suggest that these results be presented from the beginning as only exploratory in nature. No causation should be implied, only relationships between variables should be presented. I fear that if the study is published as it is currently presented, that readers could make inappropriate conclusions. The exploratory nature of the study needs to be driven home."

Responses: Given that our proposed perspective (to view behaviors as drivers of subjective recourse evaluations, although acknowledging that causality most likely goes in both directions) has not been investigated in previous studies it is (1) difficult to cite the literature you are requesting; and (2) necessary to have an exploratory aim. We find it both interesting and worthwhile – both because it makes theoretical sense and because there are no studies in this field taking such position – to treat subjective resource evaluations as dependent variables. The aim is indeed exploratory, and we have thanks to your suggestion clarified this even more in this second revision. The introductory parts are substantially revised and reorganized. We are aware that causality most likely goes in both directions and provide arguments for this in the manuscript. We have changed some expressions so that results do not indicate causality, while our conclusions, given our theoretical rationale, do suggest possible directional effects. We have clarified that the correlations found between behaviors and subjective resource evaluations are interpreted given our theoretical framework. We also admit that we cannot firmly establish causality given our analyses and suggest that future research investigates this more carefully.

Comments:

"In addition, the writing has been improved, though there are still improvements to be made in the structure and word choices. A writing professional may help improve clarity and flow."

Responses: We have tried to improve language even more. Given the short time frame for the revision we have not been able to employ professional support but plan to do so if the journal editors decide to request us to proceed further with the manuscript. 

Reviewer 2 Report

I will suggest the authors to provide more evidences for the suitability of boundary range (1 to 2) for the measurement of Gender and Grocery shopping last time.

Author Response

Comments by and responses to Reviewer 2

Comments:

"I will suggest the authors to provide more evidences for the suitability of boundary range (1 to 2) for the measurement of Gender and Grocery shopping last time."

Responses:

We have never seen studies that do not measure gender binary; this is common practice. If you use a 0-1, a 1-2, or a 9-10 scale does not matter, as long as you know how to interpret the regression coefficients. If you think of including those who do not want to answer this question or do not feel comfortable with the binary gender scale (only very few of the respondents), and have a separate category for those respondents, the results would be skewed.

With the variable grocery shopping last time we tried to capture whether (2) or not (1) respondents bought eco-labelled groceries. In the survey we had to include the (third) option of never buying the product in question. As we explain in the revised manuscript (see this section pasted below) it would make no sense to include all three possible answers as variable values, since it would imply a mix of two different logics; “pro-environmental behavior in terms of buying eco-labelled products and pro-environmental behavior more generally”.Given that the variable is an index variable consisting of answers on eight items the scale is not dichotomous, even though it ranges from 1 to 2. Again, we could have made it range differently, without any statistical implications, other than it is important to know how to interpret the regression coefficients. However, we chose to stick with a range 1-2, mirroring the behaviors “not buying eco-labelled” and “buying eco-labelled products”.

Pasted from the revised manuscript:

“Grocery shopping last timewas measured by asking respondents to think about the last time they bought milk/dairy products, fruits/vegetables, eggs, flour/grain/cereals/muesli, meat/fish, canned food, oil, and coffee/tea, and indicate whether they bought an eco-labeled product or not. Respondents had the possibility to answer that they never buy the product in question. Buying non-eco-labeled (i.e., conventional) products equaled 1, buying eco-labeled products equaled 2, indicating that the higher value the more environmentally friendly behavior. It could be debated whether the third option (3), never buying a product (which is a necessary option for respondents), is part of the same scale. For instance, never buying eggs (because you never eat eggs) could in one way be viewed as the most pro-environmental behavior (it implies less environmental impact than buying whichever kind of eggs), but it would be to mix different logics; pro-environmental behavior in terms of buying eco-labelled products and pro-environmental behavior more generally. What we tried to capture here was the former. Therefore, answers 3 on these items were recoded as the mean value of the responses on the scale 1-2. The eight items were then averaged into an index variable. Hence, an interval scale ranging from 1 to 2 was created (Cronbach’s Alpha 0.756).”